# An Original ELISA-Based Multiplex Method for the Simultaneous Detection of 5 SARS-CoV-2 IgG Antibodies Directed against Different Antigens

**DOI:** 10.3390/jcm9113752

**Published:** 2020-11-21

**Authors:** Constant Gillot, Jonathan Douxfils, Julie Cadrobbi, Kim Laffineur, Jean-Michel Dogné, Marc Elsen, Christine Eucher, Sabrina Melchionda, Élise Modaffarri, Marie Tré-Hardy, Julien Favresse

**Affiliations:** 1Department of Pharmacy, Namur Research Institute for Life Sciences, University of Namur, B-5000 Namur, Belgium; constant.gillot@unamur.be (C.G.); jean-michel.dogne@unamur.be (J.-M.D.); marie.trehardy@unamur.be (M.T.-H.); j.favresse@labstluc.be (J.F.); 2Central Laboratory Department, Qualiblood sa, 5000 Namur, Belgium; sabrina.mechionda@qualiblood.eu (S.M.); elise.madaffarri@qualiblood.eu (É.M.); 3Department of Laboratory Medicine, Clinique St-Luc Bouge, 5004 Namur, Belgium; julie.cadrobbi@SLBO.be (J.C.); kim.laffineur@SLBO.be (K.L.); marc.elsen@SLBO.be (M.E.); christine.eucher@SLBO.be (C.E.); 4Department of Laboratory Medicine, Iris Hospitals South, 1060 Brussels, Belgium; 5Faculty of Medicine, Université libre de Bruxelles, 1050 Brussels, Belgium

**Keywords:** COVID-19, SARS-CoV-2, multiplex, serology, kinetics

## Abstract

Strategies to detect SARS-CoV-2 are increasingly being developed. Among them, serological methods have been developed. Nevertheless, although these may present an interesting clinical performance, they are often directed against only one antigen. This study aims at evaluating the clinical performance of an innovative multiplex immunoassay (i.e., CoViDiag assay) detecting simultaneously the presence of antibodies directed against N, S1, S2, RBD and NTD antigens. Sensitivity was evaluated in 135 samples obtained from 94 rRT-PCR confirmed coronavirus disease 2019 (COVID-19) patients. Non-SARS-CoV-2 sera (*n* = 132) collected before the COVID-19 pandemic with potential cross-reactions to the SARS-CoV-2 immunoassay were included in the specificity analysis. The antibody signature was also studied in hospitalized and non-hospitalized patients. The specificity of the CoViDiag assay was excellent for all antibodies (99.2 to 100%) using adapted cut-offs. None of the false positive samples were positive for more than one antibody. The sensitivity obtained from samples collected 14 days since symptom onset varied from 92.0 to 100.0% depending on the antibody considered. Among samples collected more than 14 days after symptom onset, 12.8, 66.3, 3.5, 9.3, 5.8 and 2.3% were positive for 5, 4, 3, 2, 1 or 0 antibodies, respectively. A trend toward higher antibody titers was observed in hospitalized patient in the early days since symptom onset. However, no significant difference was observed compared to non-hospitalized patients after 14 days since symptom onset. The clinical performance of the CoViDiag 5 IgG assay is sufficient to recommend its use for the detection and the characterization of the antibody signature following SARS-CoV-2 infection. The combination of several antigens in the same test improves the overall specificity and sensitivity of the test. Further research is needed to investigate whether this strategy may be of interest to identify severe disease outcome in patients with SARS-CoV-2 infection.

## 1. Introduction

The Severe Acute Respiratory Syndrome Coronavirus 2 (SARS-CoV-2) is responsible for the ongoing pandemic and has led to significant morbidity and mortality. The number of confirmed cases exceeds 38 million and the number of deaths worldwide has passed the one million mark [1]. Currently, the gold standard method for the diagnosis of coronavirus disease 2019 (COVID-19) is detection of SARS-CoV-2 ribonucleic acid (RNA) in nasopharyngeal swabs through real-time reverse transcription polymerase chain reaction (rRT-PCR) [2]. However, the accuracy of rRT-PCR detection relies on several factors including samples types, sample collection, time since infection, viral load, transport and storage [2,3,4]. Additionally, the rRT-PCR is not able to detect past infection and the technique requires high workload, skillful operators, expensive instruments and crucial biosafety measures [2,5].

The detection of anti-SARS-CoV-2 antibodies represents an additional method for the diagnosis of COVID-19 [4,6]. Serological assays serve as an adjunction test in patients with symptoms suggestive of COVID-19 (i.e., >14 days since symptom onset) but with a negative rRT-PCR result [7,8]. Use of serological testing to evaluate seroprevalence, to identify convalescent plasma donors, to monitor herd immunity and for risk predictions have also been proposed [3,7]. A wide range of serology immunoassays have been developed to complement the rRT-PCR, with different SARS-CoV-2 antigen targets and formats [2,9,10,11,12,13,14,15]. The main SARS-CoV-2 antigens used are the nucleocapsid protein (N) and the spike protein (S) [10,16,17,18]. The nucleocapsid participates in RNA packaging and virus particle release; it is an internal protein. The transmembrane spike glycoprotein comprises two functional subunits responsible for binding to the host cell receptor (N-terminal S1 subunit (NTD)) and fusion of the viral and cellular membranes (C-terminal S2 subunit) [19,20]. The receptor-binding domain (RBD) is located at the C-terminal region of the S1 subunit, and assays using this protein have also been developed [21,22]. The performance of these assays varied because of the choice of the antigen for a particular target, the nature and structure of the target itself (purified vs. recombinant, full-length vs. truncated, eukaryotic vs. prokaryotic expression system), or the disparity of the patients cohorts [2,6,12,23,24].

The aim of this study was to assess the clinical performance of a multiplex assay designed to detect S1-, S2-, N-, RBD- and NTD-directed antibodies. The major promises of multiplex methods for anti-SARS-CoV-2 antibodies included increased performance in terms of sensitivity and of specificity [25,26,27,28], which is in line with the orthogonal testing algorithm proposed by the Centers for Disease Control and Prevention (CDC). The simultaneous detection of several antigens could also permit to predict the disease outcome by identifying various antibody signatures and allow to assess the immunological response to future vaccines with more objectivity [29].

## 2. Experimental Section

### 2.1. Study Design

This retrospective study was conducted on the 15 September 2020 at the University of Namur (Namur, Belgium) to assess the clinical performance of the CoViDiag^®^ IgG test kit (Innobiochips^®^, Loos, France). A total of 135 serum samples from 94 rRT-PCR positive patients were evaluated. A total of 132 serum samples collected before the COVID-19 pandemic were included to perform the specificity study. The study protocol was in accordance with the Declaration of Helsinki.

### 2.2. Sample Collection and Handling

Blood samples were collected from patients into serum-gel tubes (BD Vacutainer^®^ 8.5 mL tubes, Becton Dickinson, Franklin Lakes, NJ, USA) or lithium-heparin plasma tubes (BD Vacutainer^®^ 4.0 mL tubes, Becton Dickinson, Franklin Lakes, NJ, USA) according to standardized operating procedure and manufacturer recommendations. Samples were centrifuged for 10 min at 1885× g (ACU Modular^®^ Pre-Analytics, Roche Diagnostics^®^, Basel, Switzerland). One hundred thirty-five sera from 94 COVID-19 patients were collected from 21 March to 25 May 2020.

The study population displayed the following characteristics: 45 females and 51 males aged 24 to 93 years (mean age = 63 years). Information on the days since the onset of symptoms was retrieved from medical records. Symptoms included fever, cough, fatigue, muscle aches, chest pain or pressure, difficulty breathing or shortness of breath, headache, sore throat, diarrhea, loss of taste and loss of smell. Fever was the most frequent symptom (68.1%), followed by cough (60.4%), fatigue (58.2%), difficulty breathing (45.1%) and muscle aches (31.9%) [14]. Fifty patients were categorized as critical (hospitalization required, WHO scale >3) and 44 patients were categorized as non-critical (hospitalization not required, WHO scale 2–3). Non-SARS-CoV-2 sera with a potential cross-reaction to the SARS-CoV-2 immunoassay were collected before December 2019. Among them, 37 samples were kindly provided by the Department of Laboratory Medicine of Iris Hospitals South in Brussels (Brussels, Belgium) while the reminder came from the Department of Laboratory Medicine at the Clinique Saint-Luc (Bouge, Namur, Belgium; *n* = 58) and the Department of Pharmacy of the University of Namur (Namur, Belgium; *n* = 37). Samples have been stored in the laboratory serum biobank at −20 °C until analyses. Frozen samples were thawed one hour at room temperature on the day of the analysis. Re-thawed samples were vortexed before the analysis.

### 2.3. Analytical Procedures

The CoViDiag kit (Innobiochips^®^, Loos, France) is a qualitative miniaturized and parallel-arranged enzyme-linked immunosorbent assay (ELISA) intended for the detection of multiple IgG antibodies against SARS-CoV-2 in human serum or plasma. The wells of the microtiter plates are printed with 5 different antigens of the SARS-CoV-2 to bind the corresponding specific antibodies present in the patient sample. The antigens included the N protein (nucleocapsid protein, internal to the virus), the S1 protein (Subunit 1 of the Spike protein), the RBD domain of the S1 protein (receptor binding domain of the cell), the NTD domain of the S1 protein (N-terminal S1 subunit), the S2 protein (C-terminal S2 subunit) (Appendix A) [30]. After washing the wells to remove excess sample, an enzyme-labelled conjugate is added. This conjugate binds to the captured antibodies. In a second washing step, the unbound conjugate is removed. The complex formed by the bound conjugate is visualized by the addition of substrate: tetramethylbenzidine which gives locally an insoluble, blue-colored product. The intensity of this product is proportional to the quantity of specific antibodies in the sample. The cut-offs for positivity are proposed by the manufacturer (Table 1) and an algorithm is provided for the definition positive, negative, or borderline samples:-Sample considered positive: S1 and/or RBD and/or NTD is positive or S2 and/or N is >40 AU or S2 and/or N is positive and S1 and/or RBD and/or NTD and/or S2 and/or N is borderline or S2 and/or N is borderline and S1 and/or RBD and/or NTD and/or S2 and/or N is borderline.-Sample considered doubtful (or equivocal): S2 and/or N is positive or S1 and/or RBD and/or NTD is borderline.-Sample considered negative: all other cases not listed above.

Additionally, all samples were also tested on 3 other platforms for the detection of antibodies against SARS-CoV-2 antigens: Roche (total antibodies directed against the N antigen), Euroimmun (IgG antibodies directed against the N antigen) and DiaSorin (IgG antibodies directed against the S1/S2 antigen).

### 2.4. Clinical Specificity

One-hundred thirty-two non-SARS-CoV-2 sera were analyzed to determine the cross-reactivity and establish the specificity. Among these sera some of them were found positive for multiple antigens selected for the specificity. Potential cross-reactive samples included positive antibodies against HKU1 coronavirus (*n* = 78), positive antibodies against OC43 coronavirus (*n* = 118), positive antibodies against 229E coronavirus (*n* = 32) and positive antibodies against NL63 coronavirus (n = 19). The positivity against the four common coronaviruses was assessed using the BlueDiver COVIDOT 11 IgG assay (RUO assay; D-Tek, Mons, Belgium). Potential cross-reactive samples also included positive antinuclear antibodies (*n* = 5), anti-treponema pallidum antibodies (*n* = 3), anti-thyroid peroxidase antibodies (*n* = 3), antibodies RAI+ (search for irregular agglutinins) (*n* = 5), chikungunya antibody (*n* = 1), direct coombs (*n* = 1), hepatitis B antigen (*n* = 7), hepatitis C antibodies (*n* = 7), hepatitis E antibodies (*n* = 4), human immunodeficiency virus antibodies (*n* = 2), IgA *Chlamydia pneumoniae* (*n* = 1), IgM Borrelia + IgA *Helicobacter pylori* (*n* = 1), IgM *Chlamydia pneumoniae* (*n* = 1), IgG *Chlamydia trachomatis* (*n* = 1), IgG *Coxiella burnetii* (*n* = 2), IgM *Coxiella burneti* (*n* = 1), IgM cytomegalovirus (*n* = 13), IgM Epstein-Barr virus viral capsid (*n* = 5), IgM *Mycoplasma pneumoniae* (*n* = 6), IgM parvovirus B19 (*n* = 8), IgM *Toxoplasma gondii* (*n* = 11), influenza A antibodies (*n* = 4), influenza A and B (*n* = 1), high level of total IgG (17.40 g/L) (normal range: 7.00–16.00 g/L) (*n* = 1), both high levels of total IgM (5.26 g/L; normal range: 0.4–2.3 g/L) and total IgG (28.67 g/L) (*n* = 1), rheumatoid factor (*n* = 6), urinary infection with *Escherichia coli* (*n* = 1), urinary infection with *Klebsiella oxytoca* (*n* = 1). All these samples were collected before the COVID-19 pandemic and were stored at −20 °C.

### 2.5. Clinical Sensitivity

The 135 specimens used for the sensitivity were divided into five different groups based on days since symptom onset: 0–6 days: 23 sera; 7–13 days; 26 sera; 14–20 days; 24 sera; 21–28 days; 25 sera; and over 28 days; 37 sera. The latest sample was taken 62 days after the onset of symptoms. The evaluation of clinical sensitivity was evaluated using the cut-offs provided by the manufacturer and were optimized by ROC curve analyses.

### 2.6. Statistical Analysis

Descriptive statistics were used to analyze the data. Sensitivity was defined as the proportion of correctly identified COVID-19 positive patients initially positive by rRT-PCR SARS-CoV-2 determination in respiratory samples and with COVID-19 symptoms. Specificity was defined as the proportion of naïve patients classified as negative. A ROC curve analysis was performed to propose adapted cut-offs for each antigen to improve clinical performance. Samples included for ROC curves analyses were sera obtained from at least 14 days since symptom onset (*n* = 86) and sera from the specificity study (*n* = 132). A Mann–Whitney test was used to assess potential differences in critical and non-critical patients. Data analysis was performed using GraphPad Prism^®^ software (version 8.2.1, San Diego, CA, USA) and MedCalc^®^ software (version 14.8.1, Ostend, Belgium).

## 3. Results

### 3.1. Clinical Specificity

One-hundred and thirty-two non-SARS-CoV-2 sera were analyzed to determine the cross-reactivity and establish the specificity. When using the lower cut-off (i.e., borderline positive) provided by the manufacturer (i.e., N, S2 > 15 AU and S1, RDB, NTD > 10 AU), the specificity for the 5 different IgG antibodies varies from 92.4% (95% CI: 87.9–97.0%) to 100% [95% CI: 97.2–100%]. When considering the antigens independently, a specificity of 100% [95% CI: 97.2–100%] was obtained for IgG directed against N and S1, 97.7% (95% CI: 95.2–100%) for IgG directed against RBD and 98.5% (95% CI: 96.4–100%) for IgG directed against NTD. The lowest specificity was obtained for IgG antibodies against the S2 antigen with 92.4% (95% CI: 87.9–97.0%) (10 false positive results out of 132 samples) (Figure 1). Using the adapted cut-offs, the specificity for S2 (optimized cut-off: AU > 39), RBD (optimized cut-off: AU > 18), NTD (optimized cut-off: AU > 16) increased from 92.4 to 99.2% (95% CI: 97.7–100%), from 97.7 to 99.2% (95% CI: 97.7–100%) and from 98.5 to 99.2% (95% CI: 97.7–100%), respectively. The specificity for S1 (optimized cut-off: AU > 7) decreased from 100 to 99.2% (95% CI: 97.7–100%). The specificity for N (optimized cut-off: AU 12) remains unchanged (Figure 1). The description of false-positive samples according to the positive antigen, the AU value of the antigen concerned and the potentially cross-reactive antibody, is presented in Appendix A. None of these false positive samples were positive for more than one antigen (Figure 1).

The calculated specificities using manufacturer’s cut-off for the Roche, Euroimmun and DiaSorin assays were 100% (95% IC: 97.2–100%), 96.2% (95% IC: 92.9–99.5%) and 97.7% (95% IC: 85.9–95.9%) respectively. Using ROC-curve adapted cut-offs, the calculated specificities were 100% (95% IC: 97.2–100%), 93.9% (95% IC: 89.8–98.1%) and 90.9% (95% IC: 84.0–99.7%), respectively. The manufacturer’s cut-offs for Roche, Euroimmun and DiaSorin were ratio ≥1.0, ratio ≥0.8 and ≥12.0 AU/mL, respectively. The adapted cut-offs for Roche, Euroimmun and DiaSorin were ratio > 0.165, ratio >0.4 and >3.94 AU/mL, respectively.

If at least one antigen out five was considered sufficient to determine positivity, the specificity was 89.4% (95% CI: 84.1–94.7%) with the manufacturer’s cut-offs and 95.2% (95% CI: 91.8–99.0%) with adapted cut-offs. When using the algorithm provided by the manufacturer, the specificity was 94.7% (95% CI: 90.8–98.5%) (7 false positives out of 132 samples) (Figure 1). The manufacturer’s algorithm provided a similar specificity than the adapted cut-off (i.e., 94.7 vs. 95.4%).

### 3.2. Clinical Sensitivity

One-hundred and thirty-five samples of rRT-PCR-positive patients for SARS-CoV-2 were included in the sensitivity analyses. The sensitivity in the first week after symptom onset ranged from 0% (95% CI: 0–14.8%) (IgG direct against NTD) to 21.7% (95% CI: 3.5–40%) (IgG directed against N or S2) (Figure 2). It increased in the second week (7–13 days) with values ranging from 11.5% (95% CI: 0–25.7%) for IgG directed against NTD to 65.4% (95% CI: 45.8–85%) for IgG directed against N and S2. In the third week (14–20 days) the sensitivity peaked at 100% (95% CI: 85.7–100%) for IgG directed against S2. Positivity for IgG directed against NTD was 8.3% (95% CI: 0–18.7%). After 28 days, sensitivity increased to 100% for IgG against S2 and also increased for all other IgG against the different antigens except for IgG against N antigen, which remained lower than in week 3.

If at least one of the five positive antibodies is considered sufficient to determine positivity towards SARS-CoV-2 (a different approach than the algorithm provided by the manufacturer), the sensitivity increased (Figure 2). Maximum sensitivity ranged from 21.7% (95% CI: 3.5–40%) to 30.4% (95% CI: 10.1–50.8%) during the first week, from 65.4% (95% CI: 45.8–85%) to 76.9% (95% CI: 59.7–94.3%) during the second week, is maximal (i.e., 100%) during the third weeks, ranged from 88.0% (95% CI: 74.3–100%) to 92.0% (95% CI: 80.6–100%) during the fourth week, and finally remained maximal after 4 weeks (Figure 2). The use of the adapted cut-offs slightly reduced the global sensitivity i.e., 100% (95% CI: 90.5–100%) after 28 days using manufacturer cut-offs vs. 97.3% (95% CI: 91.8–100%) after 28 days when using the adapted cut-offs (Appendix A). According to the manufacturer’s algorithm, the sensitivity was 21.7% (95% CI: 3.5–40%) during the first week (0–6 days), 73.1% (95% CI: 54.8–91.3%) during the second week (7–13 days), 100% (95% CI: 85.7–100%) during the third week (14-20 days), 92,0% (95% CI: 80.6–100%) in the fourth week and 100% (95% CI: 90.5–100%) over 28 days (Figure 2).

On the Roche, Euroimmun and DiaSorin platforms, the calculated sensitivities beyond 14 days for assays ranged from 91.7% (95% IC: 79.7–99.9%) to 100% (95% IC: 90.5–100%), from 95.8% (95% IC: 87.2–99.9%) to 100% (95% IC: 90.5–100%) and from 95.8% (95% IC: 87.2–99.9%) to 100% (95% IC: 90.5–100%) using ROC-curve adapted-cut-offs, respectively. Using manufacturer’s cut-offs, the calculated sensitivities beyond 14 days ranged from 87.5% (95% IC: 73.2–99.9%) to 100% (95% IC: 90.5–100%), from 88.0% (95% IC: 74.3–99.9%) to 95.8% (95% IC: 87.2–100%) and from 83.3% (95% IC: 61.6–96.7%) to 94.6% (95% IC: 82.7–100%), for Roche, Euroimmun and DiaSorin, respectively.

A dynamic of increase in the different antibody titers (AU) and the positivity rate (%) has been observed over time since the onset of symptoms for all IgG antibodies (Figure 3). Among samples collected more than 14 days after symptom onset, 11 of the 86 samples were positive for the 5 antibodies (12.8%), 57 out of 86 were positive for 4 antibodies (66.3%), 3 out of 86 were positive for 3 antibodies (3.5%), 8 out of 86 were positive for 2 antibodies (9.3%), 5 out of 86 were positive for 1 antibody (5.8%) and 2 were not positive (2.3%). The highest titers of antibodies (AU) were detected for the N antigen and the lowest levels for the NTD antigen. There was a delay in the increase in IgG antibody positivity rate to RBD and S1 compared to N and S2 (Figure 3).

### 3.3. Critical Versus Non-Critical Patients

Even if we observed a trend to have higher signals for N, S1, S2 and RBD antibodies from 14 days since symptom onset in critical patients, the differences were not statistically different compared to non-critical patients in our cohort (*p* > 0.05) (Figure 4). Interestingly, higher antibody levels in hospitalized vs. non-hospitalized patients were observed in samples obtained early (0–6 days and 7–13 days) since the onset of COVID-19 symptoms (Figure 5).

## 4. Discussion

This study is the first to evaluate the clinical performance of the CoViDiag IgG test kit, which is designed to detect the presence of IgG antibodies to 5 SARS-CoV-2 antigens. It includes a total of 135 serum samples obtained from 94 rRT-PCR-positive patients and a total of 132 Non-SARS-CoV-2 serum samples. The sensitivity of the CoViDiag test kit after 14 days since symptom onset varies from 92.0 to 100.0%. However, it remains too low in the first 6 days after the onset of symptoms to correctly identify all positive patients, as confirmed by other studies [31,32]. Nevertheless, detection of early seroconversion may be useful to understand the heterogeneity of clinical presentations and the interactions between antibody isotypes and viral proteins [33,34,35]. Many of the current serological assays for the detection of SARS-CoV-2 antibodies on the market are mono-antigenic, with some of them up to two antigens [36]. Currently, most of the tests available on the market use the N antigen (Roche, Euroimmun, Abbott) [5,9,10,25], the S1 antigen (Euroimmun) [9,12,37], the combination of S1 + S2 antigens (DiaSorin) [9,12,38], some tests also use the RBD antigen (Siemens, Wantai) [21,22,39], or the combination of S and N antigens (iFlash, Maglumi, Mikrogen) [10,38]. Attempts to correlate the results obtained with different tests using the same antigens are often unsatisfactory. This demonstrates the lack of overall harmonization that exists within these tests [10,22,40,41]. In addition, most tests focus on N and S [9,16]. At present, a limited number of studies have evaluated the possibility of using the multiplex technique in the detection of anti-SARS-CoV-2 IgG antibodies. Several techniques have been used; Luminex-based assay (N and S antigens) [27], protein micro-array assay (N and S antigens) [26], bead-based immune assay (S1, RBD and N antigens) [25,26] and a solid-phase chemiluminescent assay (trimeric S, S1, RBD and N antigens) [42].

According to the manufacturer’s algorithm, the specificity of the CoViDiag was evaluated at 94.7%. The adapted cut-off permit to obtain a slightly higher specificity, i.e., 95.4%, compared to the algorithm of the manufacturer (Figure 1). These specificities were not optimal but, taken separately, the specificity of each antigen was excellent (i.e., 99.2 to 100%) when using adapted cut-offs, suggesting that improvements of the algorithm could be done (Figure 1). Additionally, none of the false positive samples were positive for more than one antigen suggesting that a sample with two positive antibodies is very unlikely to be a false positive result (Figure 1). The utilization of several antigens in one assay is in line with CDC’s recommendations about the orthogonal testing strategy [43]. Among our cohort, 14 days after symptom onset and using the adapted cut-offs, only two samples were not positive at all (2.3%) and five were positive for only one antigen (5.8%). Interestingly, for samples with only one antigen, one was positive for the N antigen and 4 were positive for the S2 antigen. As the specificity of N and S2 is 100 and 99.2% with adapted cut-offs, the risk of having a false positive is very low. The calculated specificities were similar to those obtained on Roche assay (i.e., N antigen) but higher compared to those obtained on the Euroimmun (i.e., N antigen) and DiaSorin (i.e., S1/S2 antigen) assays, whatever the use of the manufacturer’s or the adapted cut-offs.

Current data on the immune response to SARS-CoV-2 estimate that seroconversion occurs between 7 and 14 days from the onset of symptoms [4,5,14,44]. Our results confirm these observations but bring this evidence to several antigens, i.e., N, RBD, S1 and S2. When considering the individual sensitivity of the test for each antigen, the positivity rate remained low before 14 days and was higher for S2 and N antigens. After 14 days, the sensitivity increased for all antigens even though the data obtained for the NTD remains very low. The most sensitive antigens were N, S2 and RBD but interestingly, RBD is only positive when the samples were also reactive for the N and S2 antigen, except for one sample which has been reactive to NTD instead of RBD. Considering the final result of the test, i.e., positive or negative for SARS-CoV-2 according to the manufacturer’s algorithm, the sensitivity after day 14 varies between 92.0 and 100%. The adapted cut-offs obtained from the ROC curves do not significantly change the sensitivity (from 100 to 97.3%) (Figure 2 and Appendix A). These sensitivities were comparable to those obtained on Roche, Euroimmun and DiaSorin assays. The results obtained for the different antigens are in agreement with the data from the literature on other serological assays [45]. Nevertheless, the CoViDiag is able to provide in one single test information on all these antigens, dispensing the use of other analyses to confirm each positive patient and is relevant for seroprevalence studies. Amongst others, this approach might improve the turnaround time and decrease overall costs in a routine testing workflow. Furthermore, because most vaccines will use the S protein or S-domains as immunogen, [46] assays targeting the N-protein would therefore not be a good candidate to evaluate the vaccine response [47]. However, having only assays targeting the S-protein might, be misleading in some situations. If a patient develops COVID-19-related symptoms following the vaccination, an assay with multiple targets might differentiate neo-COVID-19 infection from side effects due to the vaccination (i.e., flu-like syndrome).

In our cohort, the disease severity and the antibody titers were not related in samples obtained 14 days since symptom onset (Figure 4). To et al. also reported a lack of correlation between serum SARS-CoV-2 antibody levels and clinical severity of disease [48]. Van Elslande et al. reported no significant difference in seroconversion times for IgG between critical and non-critical patients [10]. Similar results were also obtained in previous studies [49,50]. However, other research groups reported that the early detection of anti-SARS-CoV-2 antibodies, and high antibody titers, were associated to disease severity [8,39,50,51]. Interestingly, in our study, higher mean antibody titers were observed in the hospitalized cohort compared to the non-hospitalized cohort when samples were obtained early, i.e., from 0 to 6 days and from 7 to 13 days, after the onset of COVID-19 symptoms (Figure 5). However, due to the retrospective design of this study and the heterogeneity between the hospitalized versus the non-hospitalized cohorts within these timeframes i.e., from 0 to 6 days and from 7 to 13 days, definitive conclusions cannot be drawn. Such finding deserves further investigations which currently fall outside the scope of this study.

## 5. Conclusions

This study evaluates the performance of a multiplex assay for the simultaneous detection of IgG antibodies to 5 different antigens of SARS-CoV-2. The specificity of the CoViDiag assay was excellent for all antigens (99.2 to 100%) using adapted cut-offs. Moreover, none of the false positive samples were positive for more than one antigen. The sensitivity obtained from samples collected 14 days since symptom onset varied from 92.0 to 100.0% depending on the antibody considered. The role of the NTD antigen in this test remains unclear based on our results. Nevertheless, the multiplex serological test shows a definite advantage over single gene tests. The combination of several antigens in the same test improves the overall specificity and sensitivity and permits us to counterbalance the lack of sensitivity or specificity of one of the antigens. We also observed a trend toward higher antibody levels in hospitalized patients in the first two weeks following the onset of symptoms. Non-statistical differences were, however, observed between hospitalized and non-hospitalized patients after 14 days since symptom onset. Future research and development of this multiplex approach is needed for better prediction of disease severity, for monitoring the immune response in patients who have suffered from COVID-19 and for the design of future vaccination campaigns.

## Figures and Tables

**Figure 1 jcm-09-03752-f001:**
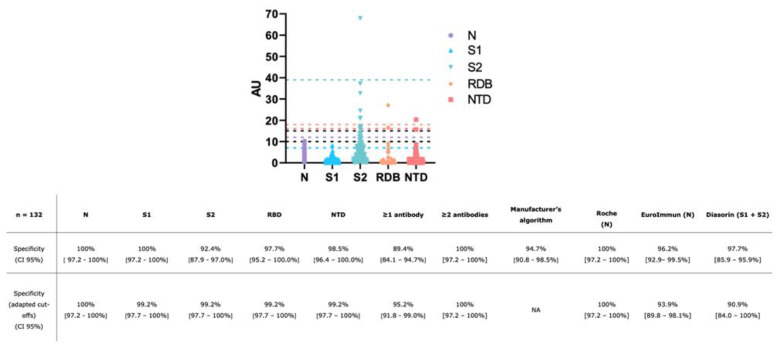
The arbitrary units obtained for each sample as a function of each antibody. Cut-offs of the manufacturer (in black; N and S2 > 15 AU and RBD, NTD and S2 > 10 AU) and optimized cut-offs (in colors) are presented on the *Y*-axis. NA: not applicable.

**Figure 2 jcm-09-03752-f002:**
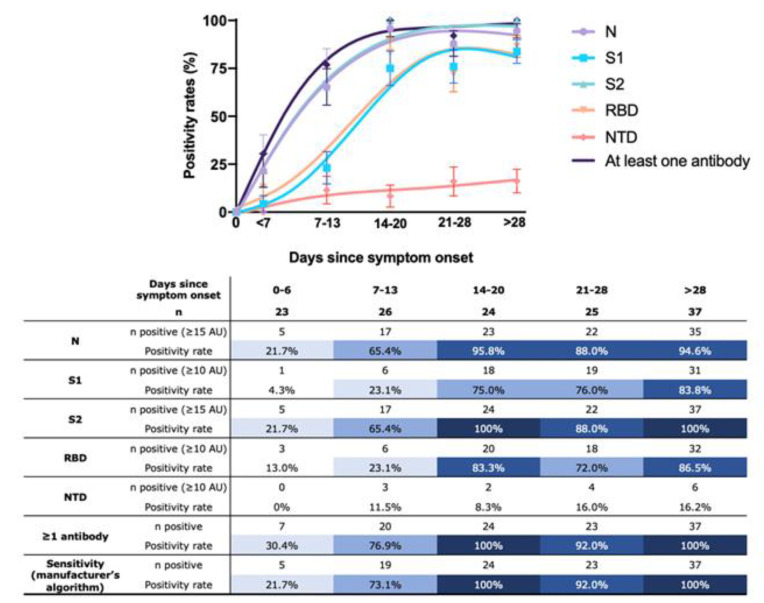
Evolution of the positivity rate according to the number of days since the onset of symptoms for each antigen (at least one positive antigen and according to the manufacturer algorithm).

**Figure 3 jcm-09-03752-f003:**
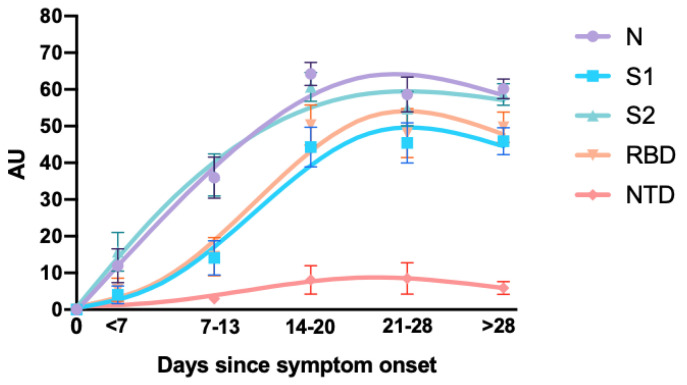
Evolution of the average of the arbitrary units obtained for each antigen in the different periods of time since the onset of the symptoms and representation of the distribution of the samples according to the number of positive antigens.

**Figure 4 jcm-09-03752-f004:**
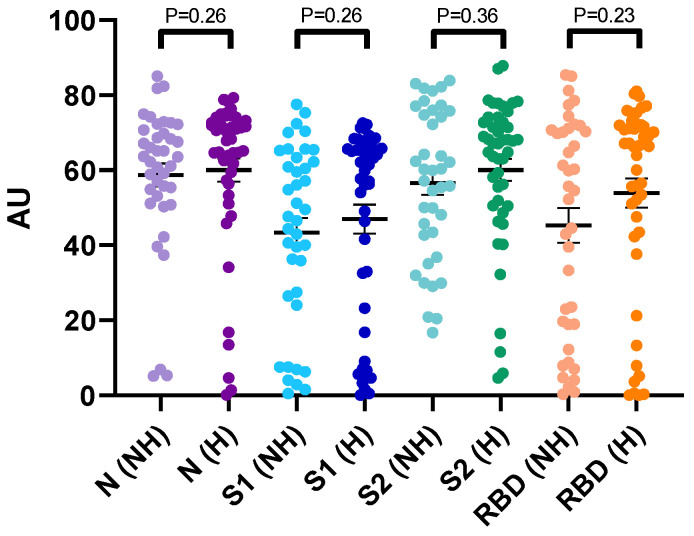
Antibody levels in hospitalized (H) and non-hospitalized (NH) patients. Samples obtained after 14 days since symptom onset were included. Purple: N-directed IgG, Blue: S1-directed IgG, Green: S2-directed IgG, Orange: RBD-directed IgG. Light colors correspond to non-hospitalized patients. Dark colors correspond to hospitalized patients.

**Figure 5 jcm-09-03752-f005:**
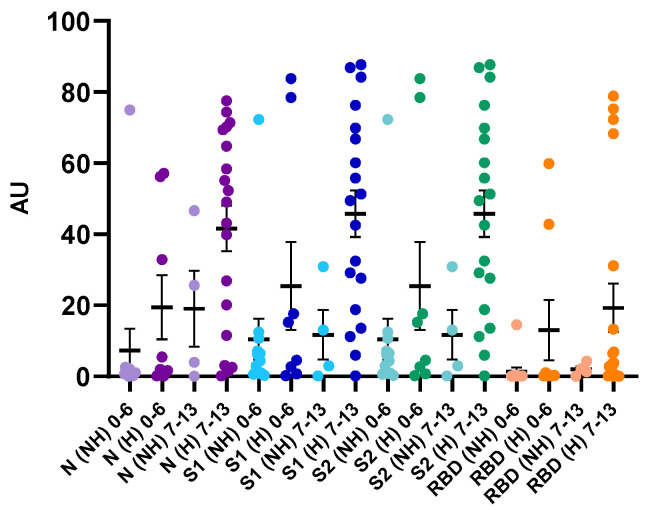
Antibody levels in hospitalized (H) and non-hospitalized (NH) patients. Samples obtained before 14 days since symptom onset were included and stratified into the first and second week. Purple: N-directed IgG, Blue: S1-directed IgG, Green: S2-directed IgG, Orange: RBD-directed IgG. Light colors correspond to non-hospitalized patients. Dark colors correspond to hospitalized patients.

**Table 1 jcm-09-03752-t001:** Negative, borderline or positive cut-offs proposed by the manufacturer for each antigen.

Antigens (AU)	Negative	Borderline	Positive	Negative (Adapted Cut-Offs)	Positive (Adapted Cut-Offs)
Nucleocapside	0–15	15–30	>30	<12	≥12
Spike S1	0–10	10–20	>20	<7	≥7
Spike S1–RBD	0–10	10–20	>20	<18	≥18
Spike S1–NTD	0–10	10–20	>20	<16	≥16
Spike S2	0–15	15–30	>30	<39	≥39

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
