# Peer review of "An Original ELISA-Based Multiplex Method for the Simultaneous Detection of 5 SARS-CoV-2 IgG Antibodies Directed against Different Antigens"

_jcm, 2020, doi:10.3390/jcm9113752_

Round 1

Reviewer 1 Report

The authors described a new multiplex method for the detection of SARS CoV-2 antibodies directed against 5 different antigens simultaneously. Compared with other methods, this multiplex method improves the overall sensitivity and specificity. Here are some minor comments that should nevertheless be taken into account:

  • Title: Authors claim that their method is "semi-quantitative" but nothing in the manuscript supports this assumption. "5 different SARS CoV-2 antibodies" is not correct and rather refer to SARS CoV-2 IgG antibodies directed against 5 different antigens. Thus title should be rephrased.
  • line 75-76: the description of the antigens used lacks some details and could be misleading for readers: N can mean N-term, Spike vs nucleocapsid proteins is not sufficiently highlighted... This should be improved.
  • Results section: For the sub-section on clinical specificity and sensitivity, a summary of the samples used should be added to make it easier for readers to understand the purpose of the experiments and the results.
  • "clinical specificity" sub-section: A very important point in assessing specificity is not only whether false positives are detected, but above all why known negative samples are detected as false positives. The authors have a lot of information on these samples and should analyse their results and offer some explanations. Is there a cross-rection with another pathogen? Is a false positive sample for more than one antigen? Authors should provide a thorough description of the false positives samples and discuss these findings.

Reviewer 2 Report

There are many COVID-19 serological assays on the market using different methods and targeting on different antigens. The current assay is interesting by testing different antigens. I would like the authors to address the following concerns:

1) Using the manufacturer's algorithm, the specificity is only 94.7%. The authors also listed the sensitivities and specificities based on different algorithms. What is the final algorithm to be recommended for clinical use? What is the assay sensitivity and specificity based on that final recommendation?

2) Although the assay is comprehensive, it is based on ELSIA which is not automated. To convenience the clinical lab to use this assay, the performance should be much better than the currently available assay. The authors should compare the performance of this assay to other commercial assays. 

Round 2

Reviewer 2 Report

The specificity of the test is too low. I don't see the clinical utility with this specificity. 

Without comparing the samples using the same samples, it is not possible to compare the sensitivity of different assays. 
